# Impact of a Playful Relaxation Intervention on Children’s Well-Being: A Mixed-Methods Study in Primary School in Portugal

**DOI:** 10.3390/healthcare14010110

**Published:** 2026-01-02

**Authors:** Sara Sarroeira, Beatriz Pereira, Guida Veiga, Wanderlei Abadio de Oliveira, José Eugenio Rodríguez-Fernández

**Affiliations:** 1Research Centre on Child Studies (CIEC), University of Minho, Campus de Gualtar, 4710-057 Braga, Portugal; sarasarroeira75@gmail.com (S.S.); beatriz@ie.uminho.pt (B.P.); 2Comprehensive Health Research Centre (CHRC), Department of Sport and Health, School of Health and Human Development, University of Évora, 7004-516 Évora, Portugal; gveiga@uevora.pt; 3School of Life Sciences, Pontifical Catholic University of Campinas, John Boyd Dunlop Avenue, Ipaussurama Garden, Campinas 13060-904, SP, Brazil; wanderleio@hotmail.com; 4Esculca Research Group, Departamento de Didácticas Aplicadas, Facultad de Ciencias de la Educación, Universidad de Santiago de Compostela, 15782 Santiago de Compostela, Spain

**Keywords:** child health, health education, child development, body awareness, first cycle of schooling, relaxation

## Abstract

**Background/Objectives**: Considering that current research highlights the role of well-being and play in children’s development and learning, and that the few publications reflecting research into relaxation methods suggest that these create conditions for well-being, the main objective of this research was to evaluate the effects of a playful intervention based on relaxation methods on the well-being of children in the first cycle of schooling. **Methods**: It is a mixed study, using quantitative and qualitative methods, with a quasi-experimental design, with an intervention group (a total of 24 sessions, based on Boski and Choque’s relaxation proposals for children) and a control group, with pre- and post-intervention assessment in both groups. Semi-structured interviews were conducted with teachers, and focus group techniques were used with children. Sixty-three children participated in this study, with an average age of 8.79 years (M = 8.79; SD = 0.676), 55.6% (35) of whom were female and 44.4% (28) male. **Results**: The results of the study indicate that the children developed passive limb relaxation and proprioceptive function, without altering their life satisfaction or aspects of emotional development. However, according to the children and teachers, the intervention developed positive emotions, bringing benefits to the classroom. **Conclusions**: This research contributes to the understanding of the effects, possibilities and potential of relaxation for children in school settings.

## 1. Introduction

Mental health and the promotion of well-being among children and adolescents have been a concern for various agents working in child development and have been increasing in recent decades [1], currently constituting a priority for several governments around the world [2]. The joint programme between the United Nations Children’s Fund and the World Health Organisation [3] on mental health and psychosocial well-being and the development of children and adolescents has made it clear that the COVID-19 pandemic and the measures taken to control it have exacerbated the mental health and well-being problems that already limited the lives of children and adolescents: depression, anxiety and behavioural disorders are identified as the main causes of illness and disability among adolescents, and suicide is the fourth leading cause of death among 15- to 19-year-olds [3]. The implementation of various strategies to promote mental health and well-being, such as life skills and social-emotional learning programmes, has been prioritised and was recently reinforced in the report Child Well-Being in an Unpredictable World [4].

The National Mental Health Plan (2007–2016) already included programmes for early childhood and school age as strategies for promoting mental health [5]. Mental health involves a positive dimension, related to psychological and physical well-being, and a negative dimension related to mental illness and psychological suffering or distress, which is defined as a syndrome that incorporates feelings of anxiety, depression, cognitive problems, irritability, anger and obsessive-compulsive behaviours [5]. Stress is a natural response to challenges and its management influences well-being [6]. In children, stress can cause symptoms such as somatic complaints, poor emotional regulation, anxiety, depression and health problems [7], and there is evidence that common mental disorders arise in adolescence, with increasing rates of anxiety and depression [4,8]. Stress, mental health and well-being are interconnected.

With regard to the conditions that enable well-being, we considered the five proposals [9]: positive emotions (feelings of happiness, including joy, pleasure and fun), engagement (absorption or connection in an activity), positive relationships (feelings of connection, security and support from others), meaning or purpose (existence with meaning or purpose, putting individual strengths at the service of something considered greater than oneself) and fulfilment (drive or ambition to achieve personal goals) [10]. The concept of subjective well-being refers to what people think and feel, integrating three associated factors: positive affect, negative affect and life satisfaction [11].

The need to disseminate effective strategies to protect children and young people from the dysfunctional effects of stress [12], equipping individuals with the necessary tools to deal with adverse circumstances [7], is the order of the day [2], with a consensus that the school environment plays a fundamental role in implementing effective measures to protect the psychological health and promote the well-being of children and young people [13,14] by incorporating prevention and early intervention programmes [15], particularly primary and universal prevention [8]. This is where interventions based on relaxation methods with playful practices can fit in.

Being a child is inseparable from playing. Recently, in a study characterising children’s subjective well-being during the pandemic, playing is associated by children with health and the absence of play is associated with feelings of longing and sadness [16], and in another study that addresses well-being from the perspective of children, well-being, recognised in feelings of happiness, appears to be associated with free play [17].

Playing is a child’s right (enshrined in Article 31 of the Convention on the Rights of the Child, 1989) and is considered a self-adaptive, self-determined and self-regulating process that gives children enormous possibilities and potential for development [18], promotes learning with joy and commitment on the part of children and facilitates the acquisition of skills that accompany children throughout their lives [19]. Play is a privileged means of learning; it is the natural activity of a child’s initiative that best reveals their holistic way of learning and allows the development of skills in all areas of development. When children play, they show signs of joy, pleasure and fun and are “engaged”, with positive emotions and engagement being two of the conditions that enable well-being [10]. In this sense, the playground should also be considered in the promotion of physical, social and emotional well-being in children, as it is a time of freedom and rest [20].

In relaxation methods intentionally adapted for children, such as Jacques Choque’s Concentration and Relaxation method [21] or Boski’s adapted Active Relaxation method [22], special emphasis is placed on the playful aspect of the activities. These methods are preventive and educational in nature, where the goal is not to treat or cure but rather to promote well-being and improve quality of life.

A literature review study aimed at characterising interventions based on relaxation methods for children in a school context, from a perspective of universal prevention of mental health-related issues and promotion of well-being, concludes that these can reduce overt symptoms of anxiety and improve autonomic functions (ability to relax); promote children’s autonomy and initiative; interactions with others; promote mental health; promote socio-emotional development; and short-term memory [23].

In general, relaxation methods focus on awareness and tonic-emotional regulation and include different techniques for breathing, neuromuscular relaxation and awareness of tonic changes related to movement and the absence thereof [24]. The techniques used in relaxation methods are considered to promote learning and emotional balance [15,22], contribute to the healthy development of children [25,26] and can positively influence the classroom environment and academic performance [27].

Considering the current recommendations regarding the promotion of the well-being of children and young people, particularly in a school context—that children need movement and to experience their bodies, and relaxation methods for children provide this experience and are promising in terms of promoting well-being—and that there are very few publications reflecting studies on interventions based on relaxation methods for children, and that it is rare in a school context from a universal perspective [23], there was interest in deepening knowledge about the contribution of a Playful Relaxation Intervention in schools to well-being.

The aim of this study was to evaluate the effects of a playful intervention based on relaxation methods on the well-being of children, presenting the results obtained for children in the first cycle of schooling.

## 2. Materials and Methods

### 2.1. Study Design

This is a mixed study, using quantitative and qualitative methods. The study has a quasi-experimental design, with an intervention group (G1) and a control group (G2). G1 underwent a playful intervention based on relaxation methods, while the control group carried out its normal routine. Four classes from the 1st cycle of schooling were grouped together, with G1 consisting of two classes and G2 also consisting of two classes. Pre- and post-intervention assessments were carried out in both groups.

At the end of the intervention, information was also collected from the participants in the intervention group using a focus group technique for the children and interviews for the teachers, following a standard interview model. Criteria were established for the formation of the two focus groups. The conversations were audio recorded and later transcribed, allowing for content analysis. All procedures were previously submitted to the ethics committee of the University of Minho (Portugal).

### 2.2. Participants

A total of 63 children attending the 1st cycle school participated in this study. They were in classes whose teachers had made inscription in a school project of relaxation. Classes in 3rd and 4th grade were invited to take part in the study (inclusion criteria: classes of 3rd and 4th grade). Thus, the intervention group consisted of two classes from the 3rd and/or 4th grade, and the control group also consisted of two classes from the 3rd and/or 4th grade. The children’s participation in the study, in addition to being authorised by their guardians, was voluntary and individually explained by the researcher. The 1st cycle school was from the Terceira island of Azores, Portugal.

The following criteria were established for participation in the focus groups: 6 participants from each class in the experimental group in the 1st cycle, with an equal number of girls and boys. A total of 12 children participated in the focus groups, 50% (6) female and 50% (6) male, 50% attending the 3rd year of schooling and 50% attending the 4th year of schooling. Interviews were conducted with the two teachers who participated in the intervention group.

### 2.3. Measures/Instruments

(a) Sociodemographic questionnaire (SQ), for sociodemographic characterisation. This questionnaire allowed us to gather sociodemographic information about the group of participants, namely regarding age, gender and year of schooling. It was developed by the research team.

(b) Strengths and Difficulties Questionnaire (SDQ-Por). The Strengths and Difficulties Questionnaire (SDQ) is a questionnaire that provides insight into the perceptions of parents and teachers regarding the socio-emotional development of children. It was developed by Goodman in 1994 [28] and was validated and translated for the Portuguese population [29]. This questionnaire consists of twenty-five items, organised into five scales, each consisting of five items, with each item having three response options (0—“Not true”, 1—“Somewhat true” and 2—“Very true”). In our study, the version for teachers was used, pre- and post-intervention in G1 and G2.

(c) Student Life Satisfaction Scale (SLSS): adapted to Portuguese students [30], is a self-administered instrument for assessing overall life satisfaction in children and adolescents aged 8–18 years. Responses are based on events in the ‘last few weeks’. For each item, there are six response options on a Likert scale (1—strongly disagree to 6—strongly agree). The items are added together to produce an overall life satisfaction index. The results of the scale range from 7 to 42. The higher the sum of the items, the higher the overall level of life satisfaction of the subject [30].

(d) Multidimensional Life Satisfaction Scale for Children (MLSSFC) [31], contains 50 items distributed across six factors. It is a self-administered instrument in which, for each item, the child marks how much they agree with it on a five-point Likert scale (1—not at all to 5—very much). The factors, or subscales, Friendship (10 items) and School (seven items) were used.

(e) Psychomotor Battery Tasks (PBT) [32]. This instrument allows the psychomotor profile of preschool and school-age children to be traced. The tasks of passivity (from the tonicity factor), immobility (from the balance factor) and self-image (from the body awareness factor) were applied. The tasks are applied with the observation of two psychomotor therapists, and the score is recorded between the values 1—imperfect, incomplete and uncoordinated performance; 2—performance with control difficulties; 3—controlled and adequate performance; and 4—perfect, harmonious and well-controlled performance (the scoring criteria are established for each task).

(f) Guide for the semi-structured interview conducted with teachers after the intervention, with the aim of gathering their opinions on the intervention and assessing any changes perceived by teachers that may have occurred in terms of well-being, among children and in the classroom.

(g) Guide (with some guiding topics) for conducting focus groups with children to gather their opinions on the intervention in which they participated. The information collected during and at the end of two intervention sessions was also used for data processing, using the following data collection method:

(h) Session evaluation questionnaire. In sessions 5 and 11, an evaluation questionnaire was used by the participating children, consisting of two closed questions (muscle relaxation of the body and emotion identified at the moment) and an open question, ‘What are your favourite exercises?’. It was developed by the team responsible for the ‘Let’s Relax’ project at the school.

### 2.4. Intervention

The programme, based on proposals by Boski [22], Choque [21] and Guillaud [25], consists of 24 sessions implemented twice a week (first facilitated by the psychomotor therapist and a second time by the head teacher) for 12 weeks. Each session lasts a maximum of 30 min. All sessions have a common structure that we call a standard session, organised into different parts: (i) “Welcome”; (ii) “Preparation”, which promotes a calm and engaged attitude, preparing participants for the relaxation experience; (iii) ‘Sequential relaxation exercises’, in which exercises are performed addressing neuromuscular relaxation techniques, awareness of tonic changes, breathing techniques and techniques for focusing attention and thought; (iv) “Return”; and (v) ‘Verbalisation’. The programme offers two versions with different exercises for pre-school education and for the first cycle of schooling. The version for the first cycle of schooling is mainly based on Boski’s proposals [22].

### 2.5. Procedures and Data Processing

As for quantitative analysis, all statistical analysis was performed using IBM SPSS Statistics (version 28, IBM Corporation, Armonk, New York, USA) with 5% significance. The internal consistency of the data collected by each of the instruments was verified using Cronbach’s alpha analysis; reasonable and good internal consistency values were found for the QSD responses [33], while good internal consistency values were found for the EMSVC responses, for both dimensions used and for both moments, pre- and post-intervention. Regarding the data resulting from the ESVE, non-admissible and unreasonable internal consistency values were found. In the case of the pre-intervention responses to the ESVE, internal consistency was relatively low for all questions, but improved substantially if item ESVE7, “My life is better than the life of most other people my age,” was excluded, and its analysis was maintained (ESVE_pre, α = 0.57; ESVE_post, α = 0.67) even increasing the probability of non-significant results. A descriptive analysis was performed using means, medians and standard deviation.

To assess whether the data followed a normal distribution, SPSS was used with the Kolmogorov—Smirnov and Shapiro—Wilk tests. For the data collected through QSD, EMSVC and BPM, statistical significance values lower than 0.05 were found, allowing us to state that the distribution of the data found differs from a normal distribution. Thus, as normality was not verified, and because we are dealing with a small sample size, non-parametric tests were used [34]. Regarding the data collected by ESVE, although for the control group the data follow a normal distribution (G2_pre, *p* = 0.487) and (G2_post, *p* = 0.056), it was found that the assumptions of normality are violated in the experimental group data, at the pre-intervention (*p* = 0.006) and post-intervention (*p* < 0.001) moments. As the distribution in G1 is not normal, and despite the normality in G2, the sample size in this group is 26 (<30), so it was decided to apply non-parametric tests [34].

The intra-group statistical comparison (dependent samples) was performed using the Wilcoxon test for G1 and G2, at pre- and post-intervention times. The pre- and post-intervention results in G1 and G2 (independent samples) were compared using the non-parametric Mann–Whitney U test [34]. The results of the difference, the change that occurred (post-intervention-pre-intervention) between the groups (G1 and G2), were also compared. In this way, we eliminated the differences that could exist between the groups at the beginning of the intervention by analysing only the differences recorded and collected between the pre-intervention and post-intervention (12 weeks). This comparison was performed using the Mann–Whitney U test.

As for qualitative analysis, the information collected through interviews and focus groups was processed using a set of methodological procedures that fall within the content analysis research technique [35]. Initially, the relevant data was identified, followed by the categorisation process [36]. The categories into which the relevant data were grouped were organised using open procedures [35], that is, the conceptual content of each category and the entire system was defined at the end of the procedures. However, the initial formulation of the categories was inspired by the topics established in the scripts. A qualitative categorical analysis was performed. The information collected during the implementation of the sessions was processed using quantitative methods (in the case of the “session evaluation” instrument).

## 3. Results

This study involved 63 children, 55.6% (35) female and 44.4% (28) male, aged between 8 and 11 years old, with an average age of 8.79 years (M = 8.79; SD = 0.676). This group of participants was divided into two groups: the intervention group (G1) and the control group (G2). The groups had similar characteristics in terms of gender and age. G1 had a total of 32 children, 59.4% (19) of whom were female and 40.6% (13) male, with an average age of 8.75 years (M = 8.75; SD = 0.568) and G2 with a total of 31 children, 51.6% (16) female and 48.4% (15) male, with an average age of 8.84 years (M = 8.84; SD = 0.78). The first cycle school was on the Terceira island of Azores, Portugal.

From the descriptive analysis of the data found by the SDQ-Por instrument, differences were found in the experimental group in terms of a decrease in difficulties after 12 weeks of intervention. It is observed that the mean and median decrease in all variables except for “Behavioural Problems”, in which the mean and median increase. In this case, the symptoms appear to worsen when analysing the mean of 0.68 (1.30) before the intervention and 0.88 (1.36) after the intervention. However, it should be noted that the values are close to zero both before and after the intervention (absence of difficulties) and, in fact, this is the variable with the most favourable values. In the case of the “Prosocial Behaviour” subscale, the mean increases in the sense of increased skills. These values can be found in Table 1.

In the control group, there is also an improvement in symptoms for all subscales except for “Emotional Symptoms”, where there appears to be a worsening of symptoms given the increase in the mean, 1.03 (1.49) before and 1.16 (1.32) after, but the median remains at 1.

Using SPSS, the Wilcoxon test was applied to compare the results of all SQD subscales, pre- and post-intervention in G1. It can be concluded that there are statistically significant differences for the “Hyperactivity” subscale (Z = 64.500, *p* = 0.020) and “Total Difficulties” (Z = 74.000, *p* = 0.028). The differences found in the descriptive analysis between the pre-intervention and post-intervention moments are not statistically significant for the subscales of ‘Emotional Symptoms’ (Z = 39.000, *p* = 0.219), “Behavioural Problems” (Z = 84.500, *p* = 0.132), “Problems in Relationships with Peers” (Z = 47.500, *p* = 0.086) and the “Prosocial Behaviour” subscale (Z = 130.500, *p* = 0.321).

When comparing the results before and after the intervention in G2, it is possible to conclude that there are statistically significant differences for the subscales “Behavioural Problems” (Z = 7.500; *p* < 0.001), “Hyperactivity” (Z = 17.000; *p* = 0.044), “Problems in Relationships with Peers” (Z = 22.000, *p* = 0.008), “Total Difficulties” (Z = 42.000, *p* = 0.03) and for the subscale “Prosocial Behaviour” (Z = 142.000 *p* = 0.002). There were no significant differences for the subscales “Emotional Symptoms” (Z = 86.500, *p* = 0.314).

When comparing the differences between the post- and pre-intervention scores between the groups (using the Mann—Whitney U test), significant differences were found in the mean scores between the groups for the subscales “Behavioural Problems” (U = 227.000, *p* < 0.001) and “Prosocial Behaviours” (U = 650.500, *p* = 0.027). In both cases, it can be seen that most of the differences in terms of symptom reduction occur in the control group.

The descriptive analysis of the data obtained by the SLSS instrument shows a slight increase in the mean and median responses, both in the experimental group and in the control group, with regard to “Life Satisfaction” after 12 weeks of intervention. These values can be found in Table 2.

Using SPSS, the Wilcoxon test was applied to compare SLSS results (excluding item 7) before and after intervention in G1 and G2. It can be concluded that the differences found in the descriptive analysis between the pre-intervention and post-intervention moments are not statistically significant in G1 (Z = 244.500, *p* = 0.078) or in G2 (Z = 719.500, *p* = 0.398).

When comparing the differences between the post- and pre-intervention scores between the groups (using the Mann–Whitney U test), we can conclude that these are not significant in the mean scores between groups G1 and G2 (U = 398.500, *p* = 0.942).

The descriptive analysis of the data found by the MLSSFC instrument shows a slight increase in the mean and median responses, both in the experimental group and in the control group, with regard to the “Friendship” dimension, after 12 weeks of intervention. With regard to the “School” dimension, there was a decrease in the mean and median for G1 after 12 weeks of intervention and a decrease in the mean for G2 after 12 weeks of intervention. These values can be found in Table 3.

It can be seen that the average satisfaction with life in the school dimension goes from 18.4 (close to the 75th percentile, and therefore higher than expected for the age) to 17.8 (which, although decreasing, remains above 17, which is the value found for the 50th percentile), still above the 50th percentile.

Using SPSS, the Wilcoxon test was applied to compare the results of the two dimensions of the MLSSFC, pre- and post-intervention in G1. It can be concluded that the differences found by the descriptive analysis are not statistically significant for the “Friendship” dimension (Z = 206.000, *p* = 0.678) but are statistically significant for the “School” dimension (Z = 53.000, *p* = 0.049), so, the decrease in satisfaction with school after 12 weeks of intervention is significant for G1 (experimental).

When comparing the results before and after the intervention in G2, it can be concluded that there are no statistically significant differences for the dimension “Friendship” (Z = 222.500; *p* = 0.656) or for the dimension “School” (Z = 104.000, *p* = 0.186).

When comparing the differences between the post- and pre-intervention scores between the groups (using the Mann–Whitney U test), we conclude that there are no significant differences in the mean scores between groups G1 and G2 for the dimension “Friendship’ (U = 483.500, *p* = 0.788), nor for the dimension “School” (U = 470.000, *p* = 0.759).

As for the results found through the Psychomotor Battery (PBT) Tasks with regard to the subfactors passivity, immobility and self-image, the descriptive analysis showed positive differences in the experimental group after 12 weeks of intervention, such as an increase in the mean in all subfactors and in the median in the subfactors passivity and immobility. Positive differences were also found in the control group with regard to the mean for all subfactors but not with regard to the median. Table 4 presents the descriptive results.

When testing the results with the Mann–Whitney U test, in order to compare G1 and G2 at pre-intervention and post-intervention moments, it is possible to conclude that at the pre-intervention moment, the differences found between G1 and G2 are not statistically significant for any of the following subfactors: “Passivity”; (U = 478.500, *p* = 0.835); “Immobility” (U = 390.000, *p* = 0.232); and “Self-image” (U = 429.500, *p* = 0.558).

After the intervention, the differences are not statistically significant for the subfactor “Immobility” (U = 409.500, *p* = 0.263), but they are statistically significant for the subfactors “Passivity” (U = 291.500, *p* = 0.003) and “Self-image” (U = 310.000, *p* = 0.005).

From the information gathered by the focus group technique, the topics included in the script were considered, and after content analysis, the categories and subcategories presented in Table 5 were determined.

### 3.1. Children’s Perceptions of Relaxation

The information gathered around the question “What does relaxing mean to you?” constitutes the first sub-theme, where information was organised to reveal that children have different views on what relaxing means: they refer to computer games (“when you play on your mobile phone, you also relax”); to sleeping (“for me, relaxing means sleeping”); everyday situations that consume little energy, in which they can point to a piece of furniture or a specific object without there being an activity (“staring at your mobile phone”, “sitting on the sofa watching television”); and, although fewer in number but extremely relevant, there are statements that refer to a positive individual experience with silence and thought (“relaxing is being alone in silence”; “being in silence, without hearing anything”; “for me, relaxing is clearing my mind of all thoughts”).

In general, they are already providing information on the second sub-theme, “how they relax”. All the responses, which are quite varied, refer to everyday activities and can be grouped as follows: (i) only with their own body (“yawning”, “when I stretch”); (ii) rest/relaxation activities (“sleeping, listening to music, watching television”); (iii) sports and specific activities (“I also relax by skating, which is the sport I do”, “There are also things just for relaxation, like taking time off and going to a place like yoga”); (iv) in a specific place (“being on the sofa…”, “I relax better in bed… because the bed is comfortable”); (v) with friends (“I also like to relax… with friends”).

When asked how they feel when they relax (third sub-theme), all responses revolved around feelings of comfort and/or well-being (“I enjoy that moment”, “It feels good”, “Comfortable”, drowsiness “I feel like sleeping”), positive emotions (“Happy”, “Friendly”, “I feel fresh air inside me, I feel happy”) and allusions to sensations that could be associated with fulfilment or inner peace (“I feel like I’m in another world”; “I’m in the world of Tam, tam, tam, tam!”; “I’m in a world that’s all pink and white and purple”). All comments were positive, referring to pleasant sensations.

### 3.2. Children’s Perceptions of the Intervention/Play Activities Experienced

As for the second theme, “children’s perceptions of the intervention/play activities experienced”, we present the information gathered through the focus group and also that gathered through the “Session evaluation questionnaire” during two of the sessions implemented.

(a) Information collected by the focus group.

When asked about the intervention carried out, “thinking about the exercises we did in the sessions, how did you feel after the exercises?”, the statements reveal the following feelings of:Comfort/well-being/pleasure (it was like “Oh, this is so good!”’, “I feel good, relaxed, clear-headed”, “…and I felt comfortable”).Positive emotions/absence of negative emotions (“happy”, “…I throw everything away, everything bad I’m thinking about, I throw it away”, “Because when I scream, it releases the bad things”).Drowsiness (“Sleepy”, “Because I feel like sleeping”).Relaxation (“I felt relaxed…”, “it made my arms and head relax a lot”, “Because it released tension…”). Some children consider that the effects are reflected in class (“I am more relaxed and pay more attention in class”, “More focused”).

When asked about their favourite exercises and why, they refer to their characteristics and the sensations they provoke in a generalised way, such as “massages”, “the puppet”, “the leaves”, “the balloon that bursts”, “throwing the leaves”, “the shower”, “the magic potion”, “the dive” and “the rag doll”, explaining “Because it released tension and made me feel comfortable” or “because it was relaxing” or “because it makes me want to sleep”, but in some cases also more specifically. These are as follows:Massages: “the massages were like ‘oh, this is so good’”.Puppet: “I feel soft”.The balloon that bursts: “the balloon because I like to lie on the floor”.Shower: “the shower because it seems like I’m dirty with bad things, and by rubbing good things, the bad things seem to come out with the dirt”.Throwing leaves: regarding this exercise, here is a small excerpt from the focus group: Child (class x) “Picking up leaves, oh… I don’t even know what to say, I LOVE to scream…”; Researcher “And why, how does it feel?”; Student: “I throw everything away, everything bad I’m thinking about, I throw it away”; Researcher: “And then how do you feel?”; Child: “Relaxed”. Or, in the case of one child (class y): “When I pick up the leaves, I breathe in, I feel more relaxed, and then when I scream, it feels like I’m letting go of the bad feelings”.

(b) Evaluation questionnaire.

This questionnaire allowed us to assess how the children felt at the end of sessions 5 and 11 in terms of muscle tension and emotions. Information was also collected on their favourite exercises in order to plan sessions 6 and 12.

For session 5, out of a total of 31 children who responded to the questionnaire, 90.32% of children (n = 28) reported feeling “soft like slime” (n = 2), 6.45% “hard as a rock” and one (3.23%) did not answer question one, “How does your body feel?”.

Of the 31 children, and with regard to question two, “How do you feel?”, the majority (51.61%, n = 16) indicated at the end of the session that they felt “happy” and “sleepy”, six children (19.35%) indicated “sleepy”, four children (12.9%) “happy”, three children (9.68%) marked “energetic”, one child (3.23%) marked “happy” and “energetic” and one child (3.23%) did not respond. Overall, the option “happy” was marked by 21 children (67.7%).

As for session 11, out of a total of 33 children who answered the questionnaire, 90.9% of children (n = 30) reported feeling “soft like slime”, two children (6.1%) answered “hard as a rock” and one (3%) did not answer question one. Regarding question two, thirteen children (39.4%) indicated that they feel “sleepy”, nine children (27.3%) indicated “happy” and “sleepy”, four children (12.1%) indicated “happy”, three children (9.1%) “energetic”, one child (3%) “happy” and “angry” and three children (9.1%) did not respond. Overall, the option “happy” was selected by fourteen children (42.4%) and “sleepy” by twenty-two children (66.7%).

### 3.3. Teachers’ Perceptions of the Intervention

From the information gathered from the teachers through interviews regarding their perceptions of the intervention, the themes in the script were considered, and after content analysis, the categories and subcategories presented in Table 6 were determined.

The teachers consider elements related to the effects on children to be most relevant, such as the development of personal skills, such as self-confidence and body awareness and knowledge, and to the classroom environment, as it provides a more comfortable and confident (secure) context.

As for the effects of the intervention, the teachers’ observations were organised as follows:At the level of the children, with regard to their motivation (‘they really like it’), their body awareness (“I see greater body awareness”), relaxation skills (“…they already have… can relax when it’s time to relax”) and personal development (“they have gained a lot of confidence in themselves”)At the level of class dynamics, both in promoting a positive classroom atmosphere (“it was super fun”, “they paid more attention”, “they are calmer (…) they have a different attitude”) but also as a resource in the sense that applying some exercises at key moments promotes calm and readiness to learn (“days when I felt they were more stressed (…) I said, OK, let’s relax a little bit”).

As for the functioning and structure of the intervention, the aspects mentioned are positive, such as the training meetings, with emphasis on the practical experience of the exercises and their structure (“we always had to do them, we colleagues, it was very important”, “it wasn’t tiring, it wasn’t a heavy thing (…) It was the right time”) and the relevance of the intervention also due to the context (“Besides, it was very good at this time, and in this situation, (…) I loved the project and so did they).

The teachers were also asked to comment on learning relaxation skills in a school environment. Analysis of the content of this information allowed it to be organised into categories and subcategories, according to Table 7.

Teachers consider learning relaxation skills to be very important due to their cross-cutting nature and, especially given the characteristics of today’s society, in which children are subjected to many stimuli, many activities, a very hectic life with no time for breaks, because these are skills that enable them to deal with frustration, anxiety and tension. They consider them to be very relevant at this age and during the first cycle of schooling, as they believe that this is the right time for learning skills and not just content.

When asked to comment on how this learning could be implemented in school, it was possible to group the information into the following subcategories: it should be compulsory learning, not subject to voluntary enrolment by the teacher, so it could be included in a subject, such as citizenship or physical education, or be part of school projects included in the timetable.

## 4. Discussion

The number of published studies on the effects of relaxation-based interventions with children, carried out in a school context, of a universal nature is very small, if not scarce [23]. To our knowledge, studies conducted in the area of relaxation with children have focused on the scientific evidence of the effects that different techniques have on children [23,24,37]. This study, in addition to contributing to the effects of an intervention based on relaxation methods in a school context, brings relevant data to the current scientific research landscape regarding the opinions of children, the main protagonists of education, and the thoughts of their teachers.

Children have opinions about what it means to relax. The first result of our study tells us that the children participating in the intervention group have different understandings of what relaxation is: most refer to how they relax by specifying relaxing situations, and a smaller but extremely relevant number make statements that refer to a positive individual experience with silence and thought, in which relaxation is understood more as a skill or competence.

Children consider that they relax with everyday activities, and their statements have been grouped as follows: (a) only with their own bodies (“yawning”, “when I stretch”); (b) with friends; (c) rest/relaxation activities; (d) physical activity play; (e) reference to a specific piece of furniture. Some of the references (those involving their own bodies and when referring to the intervention exercises) suggest that the exercises experienced have an impact on the child’s daily life. When asked how they feel when they relax, all the children in our study made references to feelings of comfort and/or well-being, positive emotions and feelings that could be associated with fulfilment or peace. This data may be relevant for comparison with future studies in this field of research. Overall, these results are in line with other studies [38] that conclude that children know how to relax, that they relax in different ways, that they do so in their daily lives, and that most children describe sedentary experiences in which they relax in the company of their parents on sofas or beds, and others in which children associate relaxation experiences with feelings of tranquillity and comfort [17].

Children describe their favourite exercises. Children identify their favourite exercises by referring to their characteristics (movement, letting go of the upper limbs, self-massage and partner massage, shouting) and to the sensations they provoke, which are associated with the release of tension, comfort and pleasant physical sensations, which seems to be in line with the previous study [17] and with exercises that reduce negative emotions and thoughts, such as guided imagery, which is related to the literature suggesting that visualisation and guided imagery techniques may be more beneficial for older children, as they are positively related to age [39].

Children recognise the benefits of relaxation exercises. Another finding indicates that, in the children’s opinion, relaxation sessions cause feelings of comfort/well-being/pleasure, positive emotions and the elimination of negative emotions, relaxation and the elimination of tension and drowsiness, results that are consistent with those found in the information observed and shared at the end of the sessions, which is that most children feel “happy”. These results are consistent with a study that reports an increase in positive mood and physical well-being in children after relaxation training [40]. The children also report that the sessions had an influence on increasing their attention and concentration in class activities.

The results, based on the children’s responses, are consistent with those reported by the teachers, who consider that the most relevant aspects of the intervention are the effects they observed in the children: increased body awareness and knowledge and improved relaxation skills, which confirms the literature on relaxation for children [22,25]. In fact, statistically significant differences were found for the subfactors “passivity” (U = 291.500, *p* = 0.003) and “self-image” (U = 310.000, *p* = 0.005) when comparing G1 and G2, pre- and post-intervention, making it possible to state that the intervention improved the passive relaxation capacity of the limbs, as well as the proprioceptive function in the immediate space surrounding the body.

In line with current studies [37], teachers mentioned as relevant effects those related to the development of children’s interpersonal and personal skills (such as self-confidence); related to being and acting (in the sense of calm and tranquillity); self-regulation skills [41]; and participation in the group, along with the motivation that the sessions generated in the children and the well-being it provided them [40]. The teachers also consider that the intervention has an effect on the dynamics of the class (as a resource and in promoting a positive classroom atmosphere, by providing a context of greater ease and confidence), which is consistent with the results of other studies on the importance of applying relaxation techniques in school [27].

The apparent divergence between the qualitative reports of these benefits and the limited or non-specific quantitative effects warrants careful interpretation. Contrary to expectations, the differences found in socio-emotional development are not statistically significant and cannot be attributed to the intervention. No significant differences were found for life satisfaction (U = 398.500, *p* = 0.942), satisfaction with friends (U = 483.500, *p* = 0.788) and satisfaction with school (U = 470.000, *p* = 0.759). This result does not confirm the literature, which, although scarce, has been gaining strength and affirming the positive effects of relaxation techniques in reducing emotional problems in the socio-emotional development of children [24,37,42]. It is more in line with the results of other studies suggesting that although children learn to relax physiologically and mentally, the effects of this relaxation on an emotional and behavioural level are more difficult to establish [39].

One explanation for these results may be related to the initial values found for emotional symptoms, behavioural problems, hyperactivity, problems relating to peers and prosocial behaviour, and, in the case of primary school children only, life satisfaction, satisfaction with friends and satisfaction with school, which corresponded to average values or higher than expected for this age group. Such favourable baseline levels may have produced ceiling effects, limiting observable change.

One explanation for these results may be related to the initial values found for emotional symptoms, behavioural problems, hyperactivity, problems relating to peers and prosocial behaviour, which correspond to average values or higher than expected for this age group. Such favourable baseline levels may have produced ceiling effects, limiting observable change. Another explanation is that there is a high probability that the measurement error is high or is not valid for life satisfaction, satisfaction with friends and satisfaction with school, bearing in mind that the internal consistency of the scale was relatively low. One more explanation may be the fact that it is a very small sample, reducing statistical power and increasing sensitivity to skewed distributions. Note, for example, that in the case of the hyperactivity subscale, for G1, the symptoms appear to worsen when the mean is analysed (2.95 (2.46) at the pre-test and 3.22 (2.68) at the post-test), but when the median is analysed, there is an improvement ((Me = 3) at the pre-test and (Me = 2) at the post-test), that is, the data are skewed by the mean.

Results were also obtained from the thoughts of the participating teachers, who attributed great importance to learning relaxation skills at school due to their cross-cutting nature; due to the skills they enable, such as the ability to relax, promote a state of calm and increase attention span in proposed activities, as well as skills to deal with frustration, anxiety and tension; because of the characteristics of agitation and attention difficulties present in many children; because of the characteristics of today’s society in which children are subject to many stimuli, many activities and a very hectic life with no time for breaks; and because they are relevant to these age groups and school cycles.

The importance given by teachers to the skills worked on is in line with the findings in recent studies [14], in which all actors in the school ecosystem attach greater importance to psychological health and well-being than to knowledge. Teachers’ recommendations for universal provision and curricular integration (e.g., citizenship education or physical education) are supported by similar studies [27] and align with broader calls for systematic school-based well-being promotion [9,14,43].

## 5. Conclusions

With this study, we set out to evaluate the effects of a playful intervention based on relaxation methods on the well-being of children in the first cycle of schooling. We can conclude that the intervention developed the passive relaxation capacity of the limbs and their distal extremities (hands and feet) and the proprioceptive function in the immediate space surrounding the body. We also concluded that no differences attributable to the intervention were found in aspects related to subjective well-being (such as life satisfaction, satisfaction with friends and satisfaction with school) or related to socio-emotional development.

However, based on the opinions of the children themselves and their teachers, we can conclude that the intervention promotes positive emotions, engagement and positive relationships, which are conditions that enable well-being, promote children’s development (knowledge and body awareness, ability to relax and socio-emotional development); brings benefits to the classroom (increased attention and positive atmosphere) and have a multiplier effect. We conclude, based on the children’s opinions, that among the characteristics of their favourite activities are movement, physical contact and guided imagery. Based on the thoughts of the teachers participating in the study, we can also conclude that learning relaxation skills is understood as very important and relevant, given its cross-cutting nature, the characteristics of today’s society and the skills it develops, which enable children to deal with frustration, anxiety and tension, and that its learning should be compulsory.

These results contribute to future research focusing on universal interventions in schools, in the context of promoting health and well-being using relaxation methods and techniques. We believe that the existence of interventions based on relaxation methods in a school context, within a universal primary prevention approach, would contribute to a health-promoting school, creating positive conditions for the child’s integral development and promoting life skills.

Knowing how to deal with stress will be an essential skill for today’s children, who are part of a generation that will be strongly challenged mentally, emotionally and physically, given the current global situation and the worsening conditions that lie ahead (climate emergency, humanitarian emergency, resource emergency, etc.). One limitation of the study relates to the size and type of sample: the small number of participants may constitute an additional factor for bias in the results, and regarding type, convenience sampling interferes with representativeness, preventing the results from being generalised to the larger population. It is suggested that future research should seek to include a larger number of participants, as well as to conduct longitudinal studies. The use of the parent version of the SDQ is also suggested.

## Figures and Tables

**Table 1 healthcare-14-00110-t001:** Descriptive results, first cycle, pre- and post-intervention for the SDQ subscales.

Subscales		Experimental Group (G1)						Control Group (G2)				
	N	Pre-Intervention Mean (SD) Me		N	Post-Intervention Mean (SD) Me		N	Pre-Intervention Mean (SD) Me N			Post-Intervention Mean (SD) Me	
ES	32	1.53 (1.68)	1.0	32	1.28 (1.53)	1.0	31	1.03 (1.49)	1.0	31	1.16 (1.32)	1.0
BP	31	0.68 (1.30)	0.0	32	0.88 (1.36)	1.0	31	1.16 (1.29)	1.0	31	0.58 (0.99)	0.0
H	32	3.13 (2.42)	3.0	32	2.59 (2.03)	2.5	31	2.74 (3.22)	2.0	31	2.23 (2.69)	1.0
PRP	32	1.16 (1.46)	1.0	32	0.78 (1.29)	0.0	31	1.23 (1.31)	1.0	31	0.55 (0.92)	0.0
TD	31	6.55 (4.69)	6.0	32	5.53 (4.39)	5.0	31	6.16 (5.77)	4.0	31	4.52 (4.98)	3.00
PSB	32	8.50 (2.01)	10.0	32	8.69 (1.48)	9.0	31	7.74 (2.59)	9.0	31	9.19 (1.62)	10.0

Key: Emotional Symptoms (ES), Behavioural Problems (BP); Hyperactivity (H); Peer Relationship Problems (PRP); Total Difficulties (TD); Prosocial Behaviour (PSB), at baseline pre-intervention and after 12 weeks (post-intervention).

**Table 2 healthcare-14-00110-t002:** Descriptive statistics for Life Satisfaction (LS) results.

	Experimental Group (G1)						Control Group (G2)					
	N	Pre-Intervention Mean (SD) Me		N	Post-Intervention Mean (SD) Me		N	Pre-Intervention Mean (SD) Me		N	Post-Intervention Mean (SD) Me	
LS	31	5.05 (0.85)	5.17	31	5.16 (0.96)	5.50	26	4.94 (0.61)	5.0	26	5.06 (0.75)	5.17

**Table 3 healthcare-14-00110-t003:** Descriptive statistics for MLSSFC dimension results.

Dimensions	Experimental Group (G1)						Control Group (G2)					
	N	Pre-Intervention Mean (SD) Me		N	Post-Intervention Mean (SD) Me		N	Pre-Intervention Mean (SD) Me		N	Post-Intervention Mean (SD) Me	
Friendship	30	31.1 (4.35)	32.0	30	31.3 (4.35)	32.5	29	29.41 (5.17)	31.0	29	29.55 (5.18)	30.0
School	30	18.33 (3.24)	20.0	30	17.7 (3.34)	19.0	29	17.59 (1.48)	17.0	29	16.69 (3.18)	17.0

**Table 4 healthcare-14-00110-t004:** Pre- and post-intervention descriptive results for psychomotor subfactors.

Subfactors	Experimental Group (G1)						Control Group (G2)					
	N	Pre-Intervention Mean (SD) Me		N	Post-Intervention Mean (SD) Me		N	Pre-Intervention Mean (SD) Me N			Post-Intervention Mean (SD) Me	
Passivity	29	2.86 (0.79)	3.0	29	3.72 (0.53)	4.0	31	2.90 (0.75)	3.0	31	3.16 (0.78)	3.0
Immobility	29	3.31 (0.54)	3.0	29	3.48 (0.51)	3.0	31	3.03 (0.84)	3.0	31	3.19 (0.75)	3.0
Self-image	29	3.34 (0.67)	3.0	29	3.76 (0.43)	4.0	31	3.32 (0.47)	3.0	31	3.39 (0.49)	3.0

**Table 5 healthcare-14-00110-t005:** Themes, categories and subcategories found.

Themes	Subthemes	Categories	Subcategories
1. Children’s perceptions of relaxation	1.1. General opinion	1.1.1. Associating with a playful activity/computer games 1.1.2. Individual experience of rest or relaxation 1.1.3. Individual experience with silence and thought 1.1.4. Sleeping	
	1.2. How do they relax?	1.2.1. Everyday activities	1.2.1.1. Only with their own body 1.2.1.2. Rest/relaxation activity 1.2.1.3. Physical activity play 1.2.1.4. With specific furniture 1.2.1.5. With friends
	1.3. How do they feel when they relax?	1.3.1. Feeling of pleasure, comfort and/or well-being 1.3.2. Drowsiness 1.3.3. Feeling of fulfilment or peace or connection	
2. Children’s perceptions of the intervention carried out/play activities experienced		2.1. Favourite activities	
		2.2. Effects on themselves 2.3. Effects on the class	2.2.1. Comfort/Well-being/Pleasure 2.2.2. Positive emotions/absence of negative emotions 2.2.3. Drowsiness 2.2.4. Relaxation

**Table 6 healthcare-14-00110-t006:** Themes, categories and subcategories found (interviews).

Themes	Subthemes	Categories	Subcategories
3. Educators’ perceptions of the intervention carried out	3.1. The most relevant	3.1.1. Effects on the children themselves 3.1.2. Effects on the classroom environment	
	3.2. Effects of the intervention	3.2.1. On the children 3.2.2. On classroom dynamics	3.2.1.1. Motivation 3.2.1.2. Body awareness and relaxation skills 3.2.1.3. Personal development 3.2.1.4. Interpersonal relationships 3.2.2.1. On the classroom atmosphere 3.2.2.2. Exercises as a resource
	3.3. Functioning/structure of the project	3.3.1. Training meetings 3.3.2. Relevance	

**Table 7 healthcare-14-00110-t007:** Teachers’ perceptions of learning relaxation skills at school.

Categories	Subcategories
4.1. Importance/Relevance	4.1.1. Expressions used 4.1.2. Cross-curricularity 4.1.3. Skills it enables 4.1.4. Characteristics of this teaching cycle 4.1.5. Characteristics of today’s society 4.1.6. Target age group
4.2. Implementation at school	4.2.1. Compulsory nature 4.2.2. Integration into curricula 4.2.3. School project included in timetable

## Data Availability

The original contributions presented in this study are included in the article. Further inquiries can be directed to the corresponding author.

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
