# Peer review of "Impact of a Playful Relaxation Intervention on Children’s Well-Being: A Mixed-Methods Study in Primary School in Portugal"

_healthcare, 2026, doi:10.3390/healthcare14010110_

Round 1
Reviewer 1 Report
Comments and Suggestions for Authors
Dear Authors,
I appreciate the opportunity to review your manuscript, which aim is to evaluate the effects of a playful intervention based on relaxation techniques on the well-being of children in the first cycle of schooling. I believe it is a timely contribution to the scientific community.
I congratulate you on conducting such a comprehensive study, using both quantitative and qualitative methodologies, which allows us to understand not only the teachers' perspectives but also the children's.
After reading and analyzing your document, I would like to offer the following comments/suggestions:
I believe Introduction section presents the topic in an orderly and organized manner, with a considerable number of current bibliographic references, allowing the reader to fully understand the context of the study.
In Methods section, the information provided in section 2.2 (Participants) should be included in Results section, as it describes sociodemographic characteristics of the study participants (average ages, gender, etc.). In Participants section, you should explain the recruitment procedure, inclusion and exclusion criteria, and how the participants were assigned to Control and Intervention groups… However, do not provide specific percentage figures for participants' characteristics.
In section 2.4, Intervention section, I understand that 12 modules were developed, each repeated twice, resulting in a total of 24 sessions. I suggest modifying the wording to clarify that there were 24 sessions, not 12 (perhaps by not repeating the term “session” to avoid confusion).
The Results section should begin with the sociodemographic data of the sample, which you included in section 2.2. This section should focus on presenting objective data, and the authors' interpretations of this data should be placed in the Discussion section. Therefore, it is suggested that phrases such as: “We can therefore conclude that the intervention had positive results in terms of psychomotor development: passivity and proprioception,” be removed from the Results section. The authors are congratulated for including future lines of research. The inclusion of limitations is suggested.
Thank you very much.
Author Response
Comments 1: In Methods section, the information provided in section 2.2 (Participants) should be included in Results section, as it describes sociodemographic characteristics of the study participants (average ages, gender, etc.). In Participants section, you should explain the recruitment procedure, inclusion and exclusion criteria, and how the participants were assigned to Control and Intervention groups… However, do not provide specific percentage figures for participants' characteristics.
Response 1: Thank you for pointing this out. We agree with this comment. Therefore, a new paragraph was inserted in this section. This change can be found on page 3, from line 133 to 140.
Comments 2:. In section 2.4, Intervention section, I understand that 12 modules were developed, each repeated twice, resulting in a total of 24 sessions. I suggest modifying the wording to clarify that there were 24 sessions, not 12 (perhaps by not repeating the term “session” to avoid confusion).
Response 2: Thank you for pointing this out. We agree with this comment. Changes were made on page 5, from lines 195 and 196.
Comments 3:. The Results section should begin with the sociodemographic data of the sample, which you included in section 2.2. This section should focus on presenting objective data, and the authors' interpretations of this data should be placed in the Discussion section. Therefore, it is suggested that phrases such as: “We can therefore conclude that the intervention had positive results in terms of psychomotor development: passivity and proprioception,” be removed from the Results section. The authors are congratulated for including future lines of research. The inclusion of limitations is suggested.
Response 3: Thank you for pointing this out. We have, accordingly, modified the results section. Changes can be found in page 6, from line 248 to 255 and the phrase “we…proprioception” was removed what can be confirmed in page 8, line 353. Limitations were included on page 15, line 623.
Comments 4: Quality of English Language
Response 4: Thank you for pointing this out. We have, accordingly, the manuscript has been thoroughly reviewed to improve its academic English and clarity.
Reviewer 2 Report
Comments and Suggestions for Authors
The manuscript titled "Impact of a playful relaxation intervention on children's well-being: a mixed-methods study in primary school in Portugal" addresses a highly relevant and timely topic. Considering the increasing stress levels and mental health challenges children face post-pandemic, investigating school-based interventions is commendable. The use of a mixed-methods design is a significant strength, as it allows for capturing the children's and teachers' voices, which quantitative data alone might miss. However, there are discrepancies between the quantitative and qualitative findings that need to be addressed more transparently in the discussion. The manuscript requires moderate revisions to improve the clarity of the statistical reporting and to temper the conclusions to better align with the mixed results.
-
-
The introduction provides a solid background on mental health. However, the distinction between "standard relaxation techniques" and the "playful relaxation intervention" used in this study could be elaborated. Why is the "playful" element critical for this specific age group (1st cycle)? Theoretical grounding on play as a mediator for well-being would strengthen the rationale.
-
Please check the internal consistency (Cronbach’s alpha) reporting on Page 5. The alpha of 0.57 for ESVE_pre is quite low; the implications of this low reliability should be mentioned.
-
-
-
English language usage is generally good but contains some awkward phrasing that could be smoothed out during the proofreading stage.
-
The study has merit, particularly in its qualitative insights and the psychomotor results. I recommend publication after the authors address the issues regarding the interpretation of the mixed findings and formatting.
-
Author Response
Comments 1: The introduction provides a solid background on mental health. However, the distinction between "standard relaxation techniques" and the "playful relaxation intervention" used in this study could be elaborated. Why is the "playful" element critical for this specific age group (1st cycle)? Theoretical grounding on play as a mediator for well-being would strengthen the rationale.
Response 1: Thank you for pointing this out. We agree totally with this comment. We try to abord this subject in 5th and 6th paragraph of Introduction. We made some changes to page 2, line 85.
Comments 2: Please check the internal consistency (Cronbach’s alpha) reporting on Page 5. The alpha of 0.57 for ESVE_pre is quite low; the implications of this low reliability should be mentioned.
Response 2: Thank you for pointing this out. We have, accordingly, modified text to emphasize this point. Regarding the data resulting from the Student Life Satisfaction Scale (ESVE), its internal consistency was verified through Cronbach's alpha analysis. Non-admissible and unreasonable internal consistency values were found (Pestana & Gageiro, 2008). In the case of the pre-intervention responses to the ESVE, internal consistency was relatively low for all questions, but improved substantially if item ESVE7, "My life is better than the life of most other people my age," was excluded, what we did, and its analysis was maintained (ESVE_pre, α=0.57; ESVE_post, α=0.67). The implication of using this data is that there is a high probability that the measurement is not valid (the result may say nothing about the construction that was intended to be measured) or the measurement error is high, increasing the probability of non-significant results.
We revise and include what was suggested on page 5, line 212 and page 14, line 575.
Comments 3: English language usage is generally good but contains some awkward phrasing that could be smoothed out during the proofreading stage.
Response 3: Thank you for pointing this out. We have, accordingly, the manuscript has been thoroughly reviewed to refine the English of the manuscript's writing.
Comments 4: The figures and tables could be improved. They should be clearer and better presented.
Response 4: Thank you for pointing this out. We have, accordingly, the tables have been reviewed and adapted to APA style guidelines. In this case, there were 3 tables that did not comply with this requirement and they have been adapted, as have the others in the manuscript, thus providing the consistency requested by the reviewer.